# Health checks for adults with intellectual disability and association with survival rates: a linked electronic records matched cohort study in Wales, UK

Natasha Kennedy,[1] Jonathan Kennedy,[1] Mike Kerr,[2] Sam Dredge,[3]
Sinead Brophy  [4]

[1]National Centre for Population Health and Wellbeing, Swansea University, Swansea, UK
[2]School of Medicine, Cardiff University, Cardiff, UK
[3]School of Medicine, Swansea University, Swansea, UK
[4]School of Medicine, University of Swansea Institute of Health Research, Swansea, UK

**Correspondence to**
Dr Sinead Brophy;
s.brophy@swansea.ac.uk

## ABSTRACT

**Objective** To examine if mortality rates are lower in people with intellectual disability who have had a health check compared with those who have not had health checks.

**Setting** General practice records of 26 954 people with an intellectual disability in Wales between 2005–2017, of which 7650 (28.4%) with a health check were matched 1:2 with those without a health check.

**Primary outcome measure** Office of National Statistics mortality data; a Cox regression was utilised to examine time to death adjusted for comorbidities and gender.

**Results** Patients who had a health check were stratified by those who (1) had a confirmed health check, that is, Read Code for a health check (n=7650 (28.4 %)) and (2) had no evidence of receiving a health check in their medical record. Patients with a health check were matched for age at time of health check with two people who did not have a health check. The health check was associated with improved survival for those with autism or Down's Syndrome (HR 0.58 (95% CI 0.37 to 0.91) and HR 0.76 (95% CI 0.64 to 0.91), respectively). There was no evidence of improved survival for those diagnosed with diabetes or cancer. The people who had a health check were more likely to be older, have epilepsy and less likely to have autism or Down's syndrome.

**Conclusions** Health checks are likely to influence survival if started before a person is diagnosed with a chronic condition, especially for people with autism or Down's syndrome.

## Strengths and limitations of this study

► These findings are based on a total population cohort in one country of people with intellectual disabilities.
► Patients who have had a health check with no record in their general practitioner data will be misclassified, consequently, the differences observed will likely be higher than those reported in this study.
► Patients who live longer are more likely to have a health check (survival bias); to account for this the cohort has been paired with age matched controls.
► However, fewer controls were available to match for older people who had a health check, this could have influenced the findings for comparisons at older ages.

## INTRODUCTION

People with an intellectual disability experience more health conditions such as; epilepsy,[1] autism and dental problems.[2 3] In addition, prior research[4] observed that people with an intellectual disability are at a higher risk of leading sedentary lives and becoming overweight; subsequently developing diabetes, cardiovascular disease and respiratory disease.[5] Furthermore, this research has indicated that they are more likely to be exposed to poverty, poor housing conditions, unemployment and other social determinates of poor health.[6] Finally, communication and ability to act on health promotion information means access to healthcare provision will be reduced for people with intellectual disability. Inequalities in health are apparent for people with intellectual disabilities[1]; health checks have been recommended as one component of international health policy to address the poorer health of people with intellectual disabilities.[7] Annual health checks for people with an intellectual disability and being present on the social services register were introduced into Wales in 2006 and in England in 2007. The Cardiff/Welsh annual health check for adults is aimed at early detection and treatment.[8] The 2010 Improving Health and Lives— The Learning Disabilities Public Health Observatory review demonstrated that the health check improves detection of unmet, potentially treatable health needs.[9] Previous studies[7] further identified that health checks lead to detection

of unmet needs and targeted actions to address health needs. Additionally, a recent study indicated that general practitioners (GP) practices with Enhanced Service (eg, incentivised in offering the health check) had more health action plans and secondary care referrals.[10] The health check has been observed to improve detection of less serious health conditions such as ear wax obscuring one or both eardrums, and dental problems which will greatly influence a person's quality of life.[7] They have been found to improve detection of serious conditions such as cancer, and may improve the health promotion.[7 8 11] Conversely, it is not known if this translates into health gain. Few studies have evaluated the extent to which providing health checks leads to long term health benefits. In fact, a new evaluation of the impact of the Directed Enhanced Services in England found no significant difference between health check and controls (no health check) in terms of intermediate outcomes such as control of blood pressure.[12] In addition, annual health checks where not found to reduce emergency admissions, but they did appear to reduce preventable emergency admissions (eg, those for diabetes or Chronic Pulmonary Obstructive Disease (COPD)[12]). The lack of evidence of the long-term benefits of health checks was a justification for practices not offering them, as without this evidence the time taken to organise and undertake assessments is a substantial barrier for GPs.[13] This study specifically examines if health checks are associated with better survival and lower rates of mortality compared with those who have no health check.

## METHODS
### Study design
The Secure Anonymised Information Linkage (SAIL) databank is a data repository, which allows person-based data linkage across datasets. This databank includes Welsh GP data, hospital inpatient and outpatient records, as well as mortality data collected by the Office of National Statistics (ONS). SAIL comprises over a billion anonymised records. It uses a split-file approach to ensure anonymisation, overcome issues of confidentiality and disclosure in health-related data warehousing. Demographic data are sent to a partner organisation, National Health Service Wales Informatics Service, where identifiable information is removed. Clinical data are sent directly to the SAIL databank where an individual is assigned an encrypted Anonymised Linking Field (ALF). The ALF is utilised to link anonymised individuals across datasets, facilitating longitudinal analysis of an individual's journey through multiple health, education and social datasets. Data collected by GP's are captured via Read Codes (five-digit codes related to diagnosis, medication and process of care codes). Hospital inpatient and outpatient data are collected in the Patient Episode Database for Wales; this employs the International Classification of Diseases-10 Revision (ICD-10) clinical coding system to record clinical information regarding patients' hospital admissions,

discharges, diagnoses and operations. The ONS mortality dataset encompasses demographic data, place of death and underlying cause of death (also ICD-10).

### Patient and public involvement
This work was discussed with the Learning Disability Ministerial Advisory Group for Wales, which includes people with intellectual disabilities, and those caring for those with intellectual disabilities. Those with an intellectual disability were not involved in the design of the study or conduct of the study.

All data was selected from 2005 to 2017, patients were flagged for intellectual disability (see online supplemental file 1) and definition provided in reference 14. Patients entered the study when they became eligible for a health check; at age 18 or in 2006 whichever is later. The eligible patients were stratified by (1) ever had a confirmed health check (Read Code for a health check recorded) at any time in the patients record and (2) had no evidence of receiving a health check in their medical record.

## PROCEDURES
### Statistical analysis
STATA V.15 was used for all analysis. Descriptive statistics for each category were generated. Patients with a health check (index case) were matched for age (±5 years) at health check with two patients who did not have a health check, thus forming a matched cohort (figure 1). Cox regression was employed to examine time to mortality, from date of study entry. Analysis was adjusted for gender and comorbidities (autism with co-occurring intellectual disabilities, Down's Syndrome, diabetes, epilepsy and cancer) as these were identified as confounding variables; the cluster command in STATA was used to account for matching design. The incidence of death (deaths over number of person years follow-up) are presented for each comorbidity separately. However, the comparison patient was not matched on comorbidity in this subanalysis. People were censored if they moved out of Wales or were lost to follow-up. This dataset included everyone with an intellectual disability over the age of 18 since 2006.

## RESULTS
There were 26 954 people with an intellectual disability between 2006 and 2017 in Wales (11-year period), of these 7650 (28.4 %) have a GP record of ever having had a health check (see table 1). Consequently, this indicates that 71.6% of people with intellectual disabilities have no record of having a health check in their electronic medical records (eg, 19 304/26 954). The proportion of people who have a health check by year is between 1.69% (in 2006 when health checks began) and 12.4% in 2015 (see figure 2); subsequently, each year approximately 87.6% (eg, if at most 12.4% have a health check) of eligible people do not have a health check.

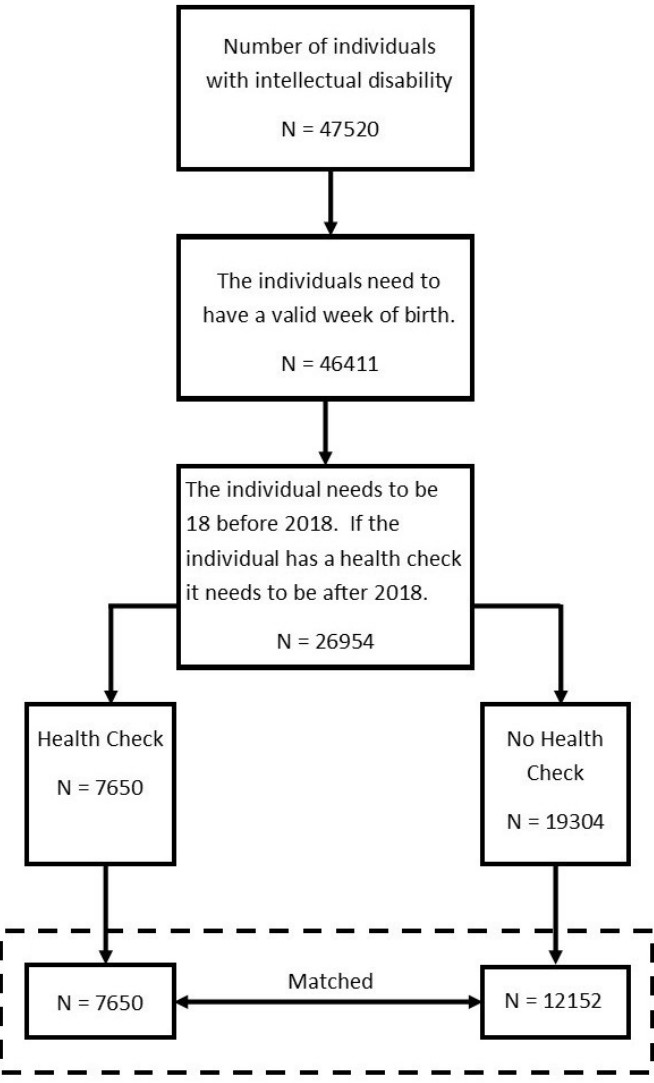

**Figure 1** Flow diagram of participant inclusion.

Patients who received a health check were more likely to be older (see table 1); the average cohort age was 40.5 years (health check) compared with 31.3 years (no health check), resulting in a difference of 9.2 years (95% CI 8.8 to 9.7). Additionally, the health check cohort were more likely to have epilepsy; 39.3% in comparison to 28.7% for the control cohort (difference 10.6% (95% CI 9.3 % to 11.9 %)). Those with autism & ID were less likely to undergo a health check (15% of those with autism have had a health check). There was no difference in socioeconomic level of those who had a health check compared with those who did not (see table 1).

Those receiving a health check were on average older than those not having health checks, in order to adjust for this survival bias, each index case was matched for the age (±5 years) at health check with two patients who had no record of undergoing a health check (see table 2). There was a trend for those who had a health check to have a lower rate of death compared with their matched comparisons (2.5 per 1000 per year fewer deaths, figure 3). Results also indicated that those with autism (HR 0.58

(95% CI 0.37 to 0.91)) and those with Down's syndrome (HR 0.76 (95% CI 0.64 to 0.91)) attained better survival rates when receiving a health check (table 2). Conversely, analysis indicated no significant increase in survival rate if a patient had diabetes, epilepsy or cancer. However, in nearly every case the diagnosis of these conditions happened before the date of the health check.

Screening for cancer was very low in both the health check and no health check groups (7.6% and 5.3%, respectively); there was no evidence of lower death rates in those who had cancer and received a health check compared with those who did not have a health check. There was a slight trend to higher rate of death in those who have a health check and have cancer. However, it should be noted that the comparison matches would have been unlikely to also have cancer.

## DISCUSSION

This study examined the medical records of 26 954 people with an intellectual disability and observed that having a health check was associated with reduced mortality for people with autism and those with Down's syndrome. Minimal evidence of reduced mortality rates was observed for those diagnosed with conditions such as diabetes or epilepsy; furthermore, no evidence was obtained to indicate that health check improved outcomes for people diagnosed with cancer.

This study only examined those who had a record of having at least one health check recorded in their medical notes with a READ code. If a person received a health check but this was not coded or recorded as such, these individuals would have been misclassified and categorised into the no health check group. Previous studies indicated that the number of people having health checks was 41% in 2008 in Wales;[15] this number was established on GP submissions to the community, primary care and the health services policy division of the Welsh Assembly Government. However, this study observed that 28.4% of people with an intellectual disability had ever received a health check coded in their medical records. This difference signifies that approximately 69% of those who might have undergone a health check had it coded in their notes, thus, the health check cohort in this study could be missing a third of the people who received a health check. Consequently, the true difference between those who have health checks and those who do not may be larger than this study has detected due to a misclassification error. In addition, this study exclusively followed individuals who are registered with their GP and had a code in their medical record identifying that they had an intellectual disability. This study could not observe the level of intellectual disability. However, the results indicated that more people who received the health check also possessed a diagnosis of epilepsy. This would suggest that those with a more severe intellectual disability are most likely to be given a health check; thus are likely to have had a higher mortality rate. This assumption is supported by prior

**Table 1** Demographics of those who have a health check compared with those with no health check (from time eligible for health check)

| | No health check | Health check | Difference (95% CI) |
|---|---|---|---|
| **Demographic characteristics** | | | |
| Total (n) | 19 304 (71.6%) | 7650 (28.4%) | |
| Average no of years of follow-up | 8.37 | 3.85 | |
| Mean age (SD) | 31.3 (17.7) | 40.5 (16.4) | −9.2 (−9.7 to −8.8) |
| **Age at baseline** | | | |
| 18–50 years old | 16 070 (83.2%) | 5539 (72.4%) | 10.8% (9.7 to 12.0) |
| More than 50 years old | 3234 (16.8 %) | 2111 (27.6%) | −10.8% (−12.0 to −9.7) |
| Male | 11 655 (60.4 %) | 4401 (57.5%) | 2.8% (1.5 to 4.2) |
| Autism | 6290 (32.6 %) | 1180 (15.4%) | 17.2% (16.1 to 18.2) |
| Down's syndrome | 3364 (17.4 %) | 915 (12.0%) | 5.5% (4.6 to 6.4) |
| Diabetes | 2332 (12.1 %) | 1030 (13.5%) | −1.4% (−2.3 to −0.5) |
| Epilepsy | 5536 (28.7 %) | 3004 (39.3%) | −10.6% (−11.9 to −9.3) |
| Cancer screening | 1030 (5.3 %) | 582 (7.6%) | −2.3% (−3.0 to −1.6) |
| Cancer | 3095 (16.0 %) | 1040 (13.6%) | 2.4% (1.5 to 3.4) |
| Died | 2527 (13.1 %) | 751 (9.8 %) | 3.3% (2.4 to 4.0) |
| **Townsend scores** | | | |
| 1 | 2515 (13.0 %) | 1087 (14.2%) | −1.2 (−2.1 to −0.3) |
| 2 | 2978 (15.4%) | 1137 (14.9%) | 0.6 (−0.4 to 1.5) |
| 3 | 3317 (17.2%) | 1535 (20.1%) | −2.9 (−3.9 to −1.9) |
| 4 | 3778 (19.6%) | 1706 (22.3%) | −2.7 (−3.8 to −1.7) |
| 5 (most deprived) | 4468 (23.1%) | 1829 (23.9%) | −0.8 (−1.9 to 0.4) |
| N/A | 2248 (11.6%) | 356 (4.7%) | 7.0 (6.3 to 7.6) |

N/A, not available.

research[12] which found those who have health checks are older, have higher levels of support needs, and are more likely to be in communal living. There were fewer controls available in the older age groups, and this means the controls were often younger than the case (within 5 years), this could influence the finding for comparisons at older ages.

The number of eligible people having a health check recorded by the GP practice each year was approximately 10%–12%. This indicates that many people may have received one health check but not undergone the health

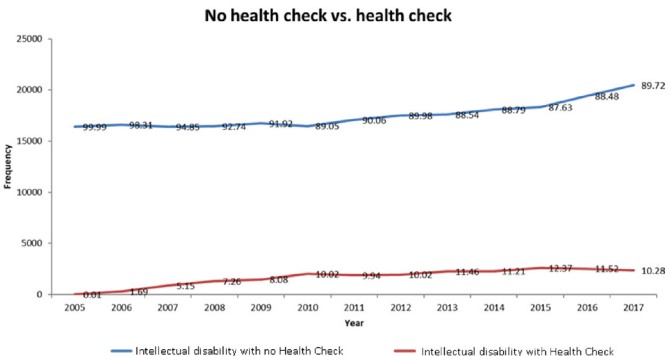

**Figure 2** Number of health checks per year.

check annually, and that the majority of people with an intellectual disability never have a health check. The study identified a lower than expected uptake of health checks in the adult population in Wales, with 71.6% having no record of a health check. While the methodology cannot identify the reason for this, this indication from the data is concerning. It appears likely that adults are not being offered health checks; either as they are not recognised as being eligible for a health check, or there is a barrier to accessing checks when offered. Further exploration of this disparity in delivery is required.

Health checks have been indicated to be valued by carers and people with an intellectual disability; they have been revealed to not increase costs, and detect health requirements earlier.[1 12] This study demonstrates that they are also associated with a long-term gain in improving survival for people with an intellectual disability, especially for those with autism or Down's syndrome. The findings of the study suggest that health checks may be most beneficial for prevention of morbidities; it did not find evidence that a health check improves survival when a person has existing health conditions, such as diabetes or cancer. It is likely that once diagnosed, those with ID have the same care for their diabetes/epilepsy/cancer

**Table 2** Survival in those who have a health check compared with no health check group (2:1 matched (for age±5 years))

| | No health check (two matches) | Health check (Index case) |
|---|---|---|
| | N=12152 | N=7650 |
| Average no of years of follow-up | 10.39 | 5.74 |
| Survival (time to death) | | |
| Total (deaths per 1000/year) | 19.6 (95% CI 18.8 to 20.3) | 17.1 (95% CI 15.9 to 18.4) |
| HR adjusted for comorbidities and gender | 1 | 0.94 (95% CI 0.87 to 1.01) |
| Males (deaths per 1000/year) | 19.0 (95% CI 18.0 to 20.1) | 16.9 (95% CI 15.3 to 18.6) |
| HR adjusted for comorbidities and gender | 1 | 0.93 (95% CI 0.83 to 1.03) |
| Females (deaths per 1000/year) | 20.2 (95% CI 19.1 to 21.4) | 17.4 (95% CI 15.6 to 19.4) |
| HR adjusted for comorbidities and gender | 1 | 0.94 (95% CI 0.84 to 1.07) |
| For those with Autism | 8.4 (95% CI 7.4 to 9.6) | 3.55 (95% CI 2.3 to 5.4) |
| Crude HR | 1 | 0.58 (95% CI 0.37 to 0.91) |
| Adjusted for comorbidities and gender | 1 | 0.56 (95% CI 0.36 to 0.89) |
| For those with Down's syndrome | 35.6 (95% CI 32.8 to 38.6) | 27.2 (95% CI 23.2 to 31.9) |
| Crude HR | 1 | 0.76 (95% CI 0.64 to 0.91) |
| Adjusted for comorbidities and gender | 1 | 0.80 (95% CI 0.67 to 0.96) |
| For those with diabetes | 31.7 (95% CI 29.4 to 34.2) | 27.6 (95% CI 23.8 to 32.2) |
| Crude HR | 1 | 0.97 (95% CI 0.81 to 1.15) |
| Adjusted for comorbidities and gender | 1 | 0.98 (95% CI 0.82 to 1.16) |
| For those with epilepsy | 24.3 (95% CI 22.9 to 25.7) | 21.9 (95% CI 19.8 to 24.2) |
| Crude HR | 1 | 1.03 (95% CI 0.91 to 1.16) |
| Adjusted for comorbidities and gender | 1 | 1.07 (95% CI 0.95 to 1.20) |
| For those with cancer screening | 25.4 (95% CI 22.6 to 28.6) | 21.2 (95% CI 17.1 to 26.3) |
| Crude HR | 1 | 1.0 (95% CI 0.77 to 1.30) |
| Adjusted for comorbidities and gender | 1 | 1.08 (95% CI 0.83 to 1.40) |
| For those with cancer | 36.6 (95% CI 34.3 to 39.0) | 38.5 (95% CI 33.7 to 43.9) |
| Crude HR | 1 | 1.18 (95% CI 1.0 to 1.4) |
| Adjusted for comorbidities and gender | 1 | 1.18 (95% CI 1.0 to 1.4) |

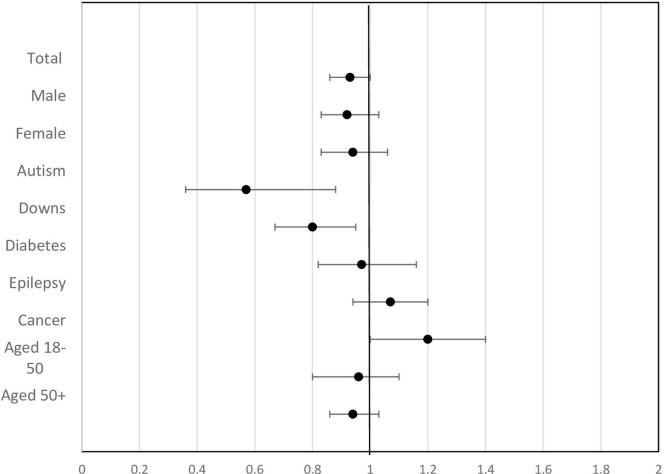

Risk ratio of survival for those having a health check compared to age/gender matched controls with no health check.

**Figure 3** Survival in those who have a health check compared with no health check group (2:1 matched (for age ±5 years).

whether receiving health checks or not. Previous research has demonstrated that emergency admission for these conditions is reduced[12] when a person undergoes health checks. The health check may improve prevention and therefore be associated with improved survival for those without existing comorbidities. There is the argument that the health check itself is not associated with improved survival; instead, this relationship is confounded by the health check being provided by more engaged GP practices,[8] more engaged family members, those with paid carers and better overall care. However, the lack of association of the health check with enhanced survival when diagnosed with a chronic condition, especially those with cancer, and a lack of association with socioeconomic deprivation found in this study, would refute this argument of confounding. Barriers and facilitators for health checks for cardiometabolic disease was reported to be more associated with the individual rather than home environment/socio economics/engagement.[16] Prior research concluded that factors such as feeling healthy, and practical issues such as the type of invitation, as well

as availability of an easy appointment acted as barriers; consequently, an argument that engaged families rather than a health check leads to benefit does not appear to be supported by the existing evidence.[16]

## CONCLUSIONS

Health checks are associated with a trend to improved survival for people with intellectual disabilities, especially for people with autism and co-occurring intellectual disabilities and Down's Syndrome. Increasing the uptake of health checks could help with prevention of morbidities and improve survival for people who do not already have chronic disease. However, there was limited evidence from this work that survival is improved when a person has existing morbidities. This study indicates benefits associated with health checks, in terms of lower rates of mortality for those with autism or Down's syndrome.

**Acknowledgements** This study uses anonymised data provided by patients, collected by the NHS as part of their care and support and held in the Secure Anonymised Information Linkage (SAIL) Databank. We would like to acknowledge all data providers that make anonymised data available for research. All data used can be accessed by contacting the SAIL gateway (https://saildatabank.com/).

**Contributors** The concept of the study and design was by MK, SB. The dataset was prepared by JK and analysis undertaken by NK with supervision from SB and MK. The findings and interpretation was after discussion with stakeholders in a consultation which was undertaken by SD. All authors contributed to drafting the paper and all have given approval of the final submission. SB acts as guarantor accepting responsiblity for the work, conduct of the study and decision to publish.

**Funding** This work was supported by Health and Care Research Wales (National Centre for Population Health and Well-being), and Health Data Research UK (NIWA1). HDR UK is funded by the UK Medical Research Council, Engineering and Physical Sciences Research Council, Economic and Social Research Council, Department of Health and Social Care (England), Chief Scientist Office of the Scottish Government Health and Social Care Directorates, Health and Social Care Research and Development Division (Welsh Government), Public Health Agency (Northern Ireland), British Heart Foundation (BHF) and the Wellcome Trust.

**Competing interests** None declared.

**Patient and public involvement** Patients and/or the public were involved in the design, or conduct, or reporting, or dissemination plans of this research. Refer to the Methods section for further details.

**Patient consent for publication** Not applicable.

**Ethics approval** Anonymised routine data analysis is not subject to ethical committee approval; it is subject to Information Governance approval. This project has been approved by the IGRP (Information Governance Review Panel) which provides independent approval that the research is within the public interest and includes representatives from the National Research Ethics Service, British Medical Association and the public.

**Provenance and peer review** Not commissioned; externally peer reviewed.

**Data availability statement** Data may be obtained from a third party and are not publicly available. All data used can be accessed by contacting the SAIL gateway (https://saildatabank.com/).

**ORCID iD**
Sinead Brophy http://orcid.org/0000-0001-7417-2858

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
