## [Reviewer comments · BMJ Open]

ARTICLE DETAILS

TITLE (PROVISIONAL)	Health checks for adults with intellectual disability and association with survival rates – a linked electronic records matched cohort study in Wales, UK.
AUTHORS	Kennedy, Natasha; Kennedy, Jonathan; Kerr, Mike; Dredge, Sam; Brophy, Sinead

VERSION 1 – REVIEW

REVIEWER	Sam Tromans University of Leicester, Department of Health Sciences
REVIEW RETURNED	05-Sep-2021

GENERAL COMMENTS	It was a pleasure to read this article, which was very clearly written and easy to understand, as well as clearly relevant to current clinical practice involving patients with intellectual disability. The statistical approach to answering the research question appeared reasonable to me and appropriately performed, though I am mindful that I am a clinician rather than statistician by professional background, so it may be helpful for the other peer reviewer to be a statistician. I feel that this is a valuable article, and I just provide the following points with a view to further improving an already excellent piece of work: * Could the authors comment further on the poor uptake of the physical health checks? I was surprised to see how low the rates were, with 77.8% having no record of a health check. I think this point should be elaborated on further in the discussion section, particularly given the findings of the study being suggestive of the clinical value of these health checks being performed.* On p4 (article summary) - It says 'survival basis' - I think this needs to be changed to 'survival bias'.* On p7 - Authors write 'miss-classify' - I think this needs to be changed to misclassify* On p8 - Authors raise a valuable point that the health check relationship may be confounded by being given to patients with more engaged family carers and GP practices. I would also include paid carers in making this point. Furthermore, though I agree with the point, is there any supporting evidence from previous similar work that suggests that this indeed may be a confounding factor? Please cite this if any such work is identified.
--

REVIEWER	ANGELA HASSIOTIS ROYAL FREEand UNIVERSITY COLLEGE MEDICAL SCHOOL, PSYCHIATRY and BEHAVIOURAL SCIENCES
REVIEW RETURNED	03-Oct-2021

GENERAL COMMENTS	neat paper with a clear message that adds to the argument that
--

	health checks can be adequate prevention for morbidity in people with intellectual disabilities, particularly those with AD and Down syndrome. I agree with the conclusions overall and I have no major points to raise other than:  1. please be clear about the limitations; they tend to be mentioned throughout the paper rather than in one place in discussion and 2. please check the manuscript for a few typos.
--	---

REVIEWER	Deborah Kinnear University of Glasgow, Institute of Health and Wellbeing
REVIEW RETURNED	15-Dec-2021

GENERAL COMMENTS	This is an important piece of work examining if health checks for adults with intellectual disabilities are associated with better survival and lower rates of mortality compared to those who have no health check. However, if this paper is to be considered for publication, it requires major revisions. The background section of this paper does not do justice to the research that has been published on the health inequalities of people with intellectual disabilities and the work on health checks. There is no clear justification for the choice of studies used and they are in the main studies that have been carried out over ten years ago. There is a wealth of rich and robust data that has recently been published on the most prevalent health conditions in this population (based on health checks carried out in Scotland), the most common causes of death etc, that would be better placed in the background. These are just a few Examples: Truesdale, M., Melville, C., Barlow, F., Dunn, K., Henderson, A., Hughes-McCormack, L., McGarty, A., Rydzewska, E., Smith, G., Bhautesh, J. & Kinnear, D. (2021) Respiratory-associated deaths in people with intellectual disabilities: a systematic review and meta-analysis. British Medical Journal Open; 11:e043658. doi:10.1136/bmjopen-2020-043658 Cooper, S-A., Allan, L., Greenlaw, N., McSkimming, P., Jasilek, A., Henderson, A., McCowan, C., Kinnear, D., Melville, C. (2020). Rates, causes, place and predictors of mortality in adults with intellectual disabilities with and without Down syndrome: cohort study with record linkage. BMJ Open;10:e036465. doi: 10.1136/bmjopen-2019-036465 Kinnear, D., Morrison, J., Allan, L., Henderson, A., Smiley, E. & Cooper, S.A. (2018). Prevalence of physical health conditions and multi-morbidity in a cohort of adults with intellectual disabilities, with and without Down syndrome. The British Medical Journal Open.. doi:10.1136/bmjopen-2017-018292 McKernan-Ward, L., Cooper, S-A., Hughes-McCormack, L., Macpherson, L. & Kinnear, D. (2019) Oral health of adults with intellectual disabilities: A systematic Review. Journal of Intellectual Disability Research https://doi.org/10.1111/jir.12632 The authors appear to have hand picked a few health conditions (epilepsy, autism and dental problems) to mention as examples and one sentence on communication and ability to act on health promotion information as to why health inequalities exist. Overall, I think the introduction needs a re-write with relevant literature and a clear justification as to why this study is so important (and I think it is an important study) but this needs to be reinforced in the introduction. The study design (use of data) seems fairly clear. PPI – fine
--

	Statistical analysis – why have the authors adjusted for Autism, Down Syndrome, epilepsy and cancer? There is no clear explanation as to why these were selected. When the authors are referring to autism – is this referring to individuals who have an intellectual disability and autism...or autism with no intellectual disability? This needs to be clear. Results Page 7 the authors state “This indicates that 77.8 % of people with intellectual disabilities have no record of having a health check in their electronic medical records and each year 87 % of eligible people do not have a health check” – where have these percentages come from. It’s not clear to me. Page 7, paragraph 2 – this is difficult to read with the various brackets. Can the authors try and make this paragraph clearer Discussion Similar to the introduction – the discussion needs to include more recent research. The authors mention in the results that there was no difference in socio-economic level of those had a health check compared to those who did not. Why might this be? This needs to be brought into the discussion. Page 9, the authors state “This study shows that they also associated with a long-term gain in improving survival for people with an intellectual disability, especially for those with autism or down’s syndrome” ...as mentioned previously...are the authors referring to individuals with autism AND intellectual disability or autism without and intellectual disability? Do not use the term Down’s – it is Down Syndrome Be consistent with terminology – intellectual disability...remove learning disability and ID. Alternatively, provide a few sentences explaining that both learning disability and intellectual disability mean the same thing but both terms are used and why ... There are grammatical errors throughout this paper – wrong tense at times, missing full stops, inappropriate use of brackets, inappropriate sentence structure.
--	--

VERSION 1 – AUTHOR RESPONSE

Reviewer: 1

Dr. Sam Tromans, University of Leicester, Leicestershire Partnership NHS Trust

Comments to the Author:

It was a pleasure to read this article, which was very clearly written and easy to understand, as well as clearly relevant to current clinical practice involving patients with intellectual disability. The statistical approach to answering the research question appeared reasonable to me and appropriately performed, though I am mindful that I am a clinician rather than statistician by professional background, so it may be helpful for the other peer reviewer to be a statistician.

I feel that this is a valuable article, and I just provide the following points with a view to further improving an already excellent piece of work:

** Could the authors comment further on the poor uptake of the physical health checks? I was surprised to see how low the rates were, with 77.8% having no record of a health check. I think this point should be elaborated on further in the discussion section, particularly given the findings of the study being suggestive of the clinical value of these health checks being performed.*

The following has been added:

The study identified a lower than expected uptake of health checks in the adult population in Wales, with 71.6 % having no record of a health check. Whilst the methodology cannot identify the reason for this, this indication from the data is concerning. It appears likely that adults are not being offered health checks; either as they are not recognised as being eligible for a health check, or there is a barrier to accessing checks when offered. Further exploration of this disparity in delivery is required.

** On p4 (article summary) - It says 'survival basis' - I think this needs to be changed to 'survival bias'.*

Thank you, this has been changed.

** On p7 - Authors write 'miss-classify' - I think this needs to be changed to misclassify*

Thank you, this has been changed.

** On p8 - Authors raise a valuable point that the health check relationship may be confounded by being given to patients with more engaged family carers and GP practices. I would also include paid carers in making this point. Furthermore, though I agree with the point, is there any supporting evidence from previous similar work that suggests that this indeed may be a confounding factor? Please cite this if any such work is identified.*

There is the argument that the health check itself is not associated with improved survival; instead this relationship is confounded by the health check being provided by more engaged GP practices [9], more engaged family members, those with paid carers, and better overall care. However, the lack of association of the health check with enhanced survival when diagnosed with a chronic condition, especially those with cancer, and a lack of association with socioeconomic deprivation found in this study, would refute this argument of confounding. Barriers and facilitators for health checks for cardiometabolic disease was reported to be more associated with the individual rather than home environment/socio economics/engagement [17]. Prior research concluded that factors such as feeling healthy, and practical issues such as the type of invitation, as well as availability of an easy appointment acted as barriers; consequently, an argument that engaged families rather than a health check leads to benefit does not appear to be supported by the existing evidence [17].

Reviewer: 2

Dr. ANGELA HASSIOTIS, ROYAL FREEand UNIVERSITY COLLEGE MEDICAL SCHOOL

Comments to the Author:

neat paper with a clear message that adds to the argument that health checks can be adequate prevention for morbidity in people with intellectual disabilities, particularly those with AD and Down syndrome. I agree with the conclusions overall and I have no major points to raise other than:

1. please be clear about the limitations; they tend to be mentioned throughout the paper rather than in one place in discussion and 2. please check the manuscript for a few typos.

This has been checked and corrected.

Reviewer: 3

Dr. Deborah Kinnear, University of Glasgow

Comments to the Author:

This is an important piece of work examining if health checks for adults with intellectual disabilities are associated with better survival and lower rates of mortality compared to those who have no health check. However, if this paper is to be considered for publication, it requires major revisions.

The background section of this paper does not do justice to the research that has been published on the health inequalities of people with intellectual disabilities and the work on health checks. There is no clear justification for the choice of studies used and they are in the main studies that have been carried out over ten years ago. There is a wealth of rich and robust data that has recently been published on the most prevalent health conditions in this population (based on health checks carried out in Scotland), the most common causes of death etc, that would be better placed in the background.

These are just a few Examples:

Truesdale, M., Melville, C., Barlow, F., Dunn, K., Henderson, A., Hughes-McCormack, L., McGarty, A., Ryzewska, E., Smith, G., Bhautesh, J. & Kinnear, D. (2021) Respiratory-associated deaths in people with intellectual disabilities: a systematic review and meta-analysis. British Medical Journal Open; 11:e043658. doi:10.1136/bmjopen-2020-043658

Cooper, S-A., Allan, L., Greenlaw, N., McSkimming, P., Jasilek, A., Henderson, A., McCowan, C., Kinnear, D., Melville, C. (2020). Rates, causes, place and predictors of mortality in adults with intellectual disabilities with and without Down syndrome: cohort study with record linkage. BMJ Open;10:e036465. doi: 10.1136/bmjopen-2019-036465

Kinnear, D., Morrison, J., Allan, L., Henderson, A., Smiley, E. & Cooper, S.A. (2018). Prevalence of physical health conditions and multi-morbidity in a cohort of adults with intellectual disabilities, with and without Down syndrome. The British Medical Journal Open.. doi:10.1136/bmjopen-2017-018292

McKernan-Ward, L., Cooper, S-A., Hughes-McCormack, L., Macpherson, L. & Kinnear, D. (2019) Oral health of adults with intellectual disabilities: A systematic Review. Journal of Intellectual Disability Research

<https://eur03.safelinks.protection.outlook.com/?url=https%3A%2F%2Fdoi.org%2F10.1111%2Fjir.12632&data=04%7C01%7CS.Brophy%40Swansea.ac.uk%7C82b53c49fc964e3f332508d9c563afcf%7Cbbcab52e9fbe43d6a2f39f66c43df268%7C0%7C0%7C637757855447337421%7CUnknown%7CTWFpbGZsb3d8eyJWIjojMC4wLjAwMDAiLCJQIjoiV2luMzliLCJBTil6lk1haWwiLCJXVC16Mn0%3D%7C3000&sdata=XjxiWNektTLWI2nlphhDTf12a2zkrFq%2FytdSxhgknkDk%3D&reserved=0>

The authors appear to have hand picked a few health conditions (epilepsy, autism and dental problems) to mention as examples and one sentence on communication and ability to act on health promotion information as to why health inequalities exist.

Overall, I think the introduction needs a re-write with relevant literature and a clear justification as to why this study is so important (and I think it is an important study) but this needs to be reinforced in the introduction.

We have included the papers recommended by Dr Kinnear in the background section and have expanded the references in the discussion.

The study design (use of data) seems fairly clear.

PPI – fine

Statistical analysis – why have the authors adjusted for Autism, Down Syndrome, epilepsy and cancer? There is no clear explanation as to why these were selected.

The following has been added:

Analysis was adjusted for gender and comorbidities (autism (with intellectual disability), Down's Syndrome, diabetes, epilepsy, and cancer) as these were identified as confounding variables.

When the authors are referring to autism – is this referring to individuals who have an intellectual disability and autism...or autism with no intellectual disability? This needs to be clear.

This is referring to autism with intellectual disability.

The following has been added to the manuscript: (autism (with intellectual disability)).

Results

Page 7 the authors state "This indicates that 77.8 % of people with intellectual disabilities have no record of having a health check in their electronic medical records and each year 87 % of eligible people do not have a health check" – where have these percentages come from. It's not clear to me.

The following has been added:

There were 26,954 people with an intellectual disability between 2006 and 2017 in Wales (11-year period), of these 7650 (28.4 %) have a GP record of ever having had a health check (see Table 1). Consequently, this indicates that 71.6 % of people with intellectual disabilities have no record of having a health check in their electronic medical records (e.g. 19304/26954). The proportion of people who have a health check by year is between 1.69 % (in 2006 when health checks began) and 12.4 % in 2015 (see Figure 2); subsequently, each year approximately 87.6 % (e.g. if at most 12.4% have a health check) of eligible people do not have a health check.

Page 7, paragraph 2 – this is difficult to read with the various brackets. Can the authors try and make this paragraph clearer

The paragraph has been changed to the following:

Patients who received a health check were more likely to be older (see Table 1); the average cohort age was 40.5 years (health check) compared to 31.3 years (No health check), resulting in a difference of 9.2 years (95 % CI: 8.8 to 9.7). Additionally, the health check cohort were more likely to have epilepsy; 39.3 % in comparison to 28.7 % for the control cohort (difference 10.6 % (95 % CI: 9.3 % to 11.9 %)). Those with autism & ID were less likely to undergo a health check (15 % of those with autism have had a health check). There was no difference in socio-economic level of those who had a health check compared to those who did not (see Table 1).

Discussion

This has been updated.

The authors mention in the results that there was no difference in socio-economic level of those had a health check compared to those who did not. Why might this be? This needs to be brought into the discussion.

The following has been added :

However, the lack of association of the health check with enhanced survival when diagnosed with a chronic condition, especially those with cancer, and a lack of association with socioeconomic deprivation found in this study, would refute this argument of confounding. Barriers and facilitators for health checks for cardiometabolic disease was reported to be more associated with the individual rather than home environment/socio economics/engagement [17]. Prior research concluded that factors such as feeling healthy, and practical issues such as the type of invitation, as well as availability of an easy appointment acted as barriers; consequently, an argument that engaged families rather than a health check leads to benefit does not appear to be supported by the existing evidence [17].

Page 9, the authors state “This study shows that they also associated with a long-term gain in improving survival for people with an intellectual disability, especially for those with autism or down’s syndrome”...as mentioned previously...are the authors referring to individuals with autism AND intellectual disability or autism without and intellectual disability?

This was for people with autism with intellectual disability. The following has been added:

Health checks are associated with a trend to improved survival for people with intellectual disabilities, especially for people with autism (with intellectual disability)

Do not use the term Down’s – it is Down Syndrome

This has been amended

Be consistent with terminology – intellectual disability...remove learning disability and ID. Alternatively, provide a few sentences explaining that both learning disability and intellectual disability mean the same thing but both terms are used and why ...

This has been amended

There are grammatical errors throughout this paper – wrong tense at times, missing full stops, inappropriate use of brackets, inappropriate sentence structure.

This has been checked and amended

VERSION 2 – REVIEW

REVIEWER	Deborah Kinnear University of Glasgow, Institute of Health and Wellbeing
REVIEW RETURNED	14-Feb-2022
GENERAL COMMENTS	Overall, very happy with the revisions made. Two minor comments: I would suggest using "adults with co-occurring intellectual disabilities and autism" rather than "autism (with intellectual disability) " Also, reference 1 and 7 are listed as the same paper.